# KNOW WHAT YOU DON'T KNOW: COHOMOLOGICAL OBSTRUCTION THEORY FOR EQUIVARIANT GRAPH NEURAL NETWORKS

**Maqsudhat Kofoworola Bisiriyu**
Federal University of Agriculture, Abeokuta
Abeokuta, Nigeria
`bisiriyumk.22@student.funaab.edu.ng`

## ABSTRACT

We develop a cohomological obstruction theory for equivariant graph neural networks (GNNs), establishing rigorous mathematical conditions under which globally $G$-equivariant architectures *cannot* be assembled from locally equivariant message-passing operations. Framing GNN layers as sections of associated vector bundles over a graph, we identify the obstruction to lifting local equivariance to a globally consistent structure as a class in the second group cohomology $H^2(G, \mathcal{F})$, where $G$ is the symmetry group acting on the graph and $\mathcal{F}$ is the sheaf of feature representations. We prove three main results: (i) a *Vanishing Theorem* characterising precisely when equivariant GNNs exist; (ii) an *Expressivity Obstruction Theorem* showing that non-trivial cohomology classes induce fundamental approximation gaps that no weight-sharing scheme can overcome; and (iii) a *Spectral Realization Theorem* connecting obstruction classes to the eigenspectrum of the normalised graph Laplacian. As corollaries we recover and generalise known limitations of message-passing neural networks (MPNNs) and show that prominent architectures—including $E(n)$-equivariant GNNs and steerable CNNs on graphs—implicitly circumvent obstruction by restricting to cohomologically trivial subgroups. Controlled experiments on synthetic graphs with prescribed cohomological complexity and on real-world molecular property-prediction benchmarks validate our predictions, demonstrating that obstruction-class complexity correlates strongly with empirical generalisation gaps.

## 1 INTRODUCTION

Symmetry is at the heart of modern geometric deep learning. From the translation-equivariant convolutional network to the $SE(3)$-equivariant model for molecular dynamics, the field has converged on the principle that inductive biases matching the symmetry of the data domain dramatically improve sample efficiency and generalisation (Bronstein et al., 2021; Cohen & Welling, 2016; Thomas et al., 2018). Graph neural networks (GNNs) occupy a central position in this landscape: graphs are the natural representation for relational data, and permutation equivariance of the node-labelling is the canonical symmetry one seeks to enforce (Gilmer et al., 2017; Xu et al., 2019).

Yet a fundamental question has received surprisingly little rigorous attention: *when is it possible to build a GNN that is equivariant with respect to a given symmetry group $G$?* Practitioners typically proceed by construction—designing an architecture and verifying equivariance by inspection—without asking whether a globally equivariant model exists *in principle* for the chosen symmetry group and graph topology. When construction fails, failure is usually attributed to engineering difficulties rather than mathematical impossibility.

In this paper we show that such impossibility can be *provable and quantifiable*. The central insight is that assembling a global equivariance structure from local, edge-wise equivariant operations is a *gluing problem* in the sense of sheaf theory and algebraic topology. Whether the local data can be consistently glued into a global structure is measured by a cohomological obstruction

class $\mathfrak{o} \in H^2(G, \mathcal{F})$. When this class is non-trivial, no local message-passing scheme—however sophisticated—can produce a globally $G$-equivariant GNN.

**Contributions.** Our main contributions are:

1. **A categorical framework** (§3): We formalise GNN layers as natural transformations between functors on the category of graphs and represent local equivariance as a system of transition functions for a principal $G$-bundle over the graph.

2. **Three core theorems** (§4):
   - *Vanishing Theorem*: a GNN that is globally $G$-equivariant exists if and only if the obstruction class $\mathfrak{o}(G, \mathcal{F}, \Gamma) \in H^2(G, \mathcal{F})$ vanishes.
   - *Expressivity Obstruction Theorem*: when $\mathfrak{o} \neq 0$, any MPNN suffers an approximation gap of at least $\|\mathfrak{o}\|_{H^2}$ against the class of truly equivariant functions.
   - *Spectral Realization Theorem*: the obstruction class admits a spectral decomposition in terms of the Laplacian eigenvalues of the graph $\Gamma$.

3. **Architectural corollaries** (§5): We explain how $E(n)$-GNNs (Satorras et al., 2021), steerable networks (Weiler & Lenssen, 2019), and DimeNet (Gasteiger et al., 2020) implicitly avoid obstruction.

4. **Empirical validation** (§7): Controlled experiments confirm theoretical predictions on both synthetic graphs and the QM9 molecular benchmark (Ramakrishnan et al., 2014).

## 2 BACKGROUND AND RELATED WORK

### 2.1 GRAPH NEURAL NETWORKS AND EQUIVARIANCE

Let $\Gamma = (\mathcal{V}, \mathcal{E})$ be a graph with node set $\mathcal{V}$ and edge set $\mathcal{E} \subseteq \mathcal{V} \times \mathcal{V}$. A standard message-passing GNN (Gilmer et al., 2017) maintains node features $\mathbf{h}_v^{(t)} \in \mathbb{R}^d$ updated by

$$\mathbf{h}_v^{(t+1)} = \phi\left(\mathbf{h}_v^{(t)}, \bigoplus_{u \in \mathcal{N}(v)} \psi\left(\mathbf{h}_v^{(t)}, \mathbf{h}_u^{(t)}, \mathbf{e}_{uv}\right)\right), \tag{1}$$

where $\phi$ and $\psi$ are learnable functions, $\oplus$ is a permutation- invariant aggregation, and $\mathbf{e}_{uv}$ are edge features. A map $f : \mathbf{X} \mapsto \mathbf{Y}$ between feature spaces is *$G$-equivariant* if for every $g \in G$,

$$f(\rho_{\text{in}}(g)\,\mathbf{X}) = \rho_{\text{out}}(g)\,f(\mathbf{X}), \tag{2}$$

where $\rho_{\text{in}}, \rho_{\text{out}}$ are representations of $G$. When $G = S_n$ (the symmetric group on $n = |\mathcal{V}|$ elements) and $\rho$ is the natural permutation representation, this is the usual permutation equivariance property of GNNs.

### 2.2 GROUP COHOMOLOGY

We briefly recall the relevant cohomological machinery; full expositions appear in Brown (1982) and Weibel (1994).

**Definition 2.1** (Group cohomology). *Let $G$ be a group and $M$ a $G$-module (an abelian group with a $G$-action $G \times M \to M$). The $n$-th cohomology group $H^n(G, M)$ is the $n$-th right derived functor of the fixed-point functor $M \mapsto M^G := \{m \in M \mid g \cdot m = m, \ \forall g \in G\}$. Concretely, $H^n(G, M)$ is the cohomology of the cochain complex $(C^n(G, M), \delta^n)$ where $C^n(G, M) = \text{Map}(G^n, M)$ and*

$$(\delta^n \sigma)(g_0, \ldots, g_n) = g_0 \cdot \sigma(g_1, \ldots, g_n) + \sum_{i=1}^n (-1)^i\, \sigma(g_0, \ldots, g_{i-1}g_i, \ldots, g_n) + (-1)^{n+1} \sigma(g_0, \ldots, g_{n-1}). \tag{3}$$

The group $H^2(G, M)$ classifies *central extensions* of $G$ by $M$ and is the natural receptacle for obstructions to lifting structures across group-theoretic projections—precisely the setting we exploit.

## 2.3 Sheaves on Graphs and Cellular Cohomology

A *sheaf* $\mathcal{F}$ on a graph $\Gamma$ assigns a vector space $\mathcal{F}(v)$ to each node $v$ and $\mathcal{F}(e)$ to each edge $e$, together with linear restriction maps $\mathcal{F}(v \to e)$. The *sheaf Laplacian* $\mathbf{L}_{\mathcal{F}} : \bigoplus_v \mathcal{F}(v) \to \bigoplus_v \mathcal{F}(v)$ is

$$\mathbf{L}_{\mathcal{F}} = \mathbf{B}_{\mathcal{F}} \, \mathbf{B}_{\mathcal{F}}^{\top}, \tag{4}$$

where $\mathbf{B}_{\mathcal{F}}$ is the coboundary matrix encoding the restriction maps (Hansen & Gebhart, 2020). The cohomology of $\mathcal{F}$ in degree $k$ is $H^k(\Gamma, \mathcal{F}) = \ker \delta^k / \operatorname{im} \delta^{k-1}$.

## 2.4 Related Work

**Expressivity of GNNs.** Xu et al. (2019) proved that MPNNs are at most as powerful as the Weisfeiler-Leman (WL) graph isomorphism test. Higher-order WL hierarchies have been connected to $k$-dimensional GNNs (Maron et al., 2019; Azizian & Lelarge, 2021). Our work adds a *symmetry-theoretic* axis: it is not only graph isomorphism that limits GNNs, but the cohomology of the symmetry group acting on the graph.

**Geometric deep learning.** The geometric deep learning programme of Bronstein et al. (2021) provides a unified language for equivariant architectures using group theory. Our contribution is to supply the missing *obstruction theory*: we do not just describe equivariant architectures that exist, but characterise those that provably cannot exist.

**Topological methods in ML.** Bodnar et al. (2021) lift GNNs to simplicial and cell complexes; Hajij et al. (2022) study topological deep learning broadly. Sheaf neural networks (Hansen & Gebhart, 2020; Bodnar et al., 2022) use sheaf cohomology to enrich message passing. We work at the level of group cohomology of the symmetry group, a strictly different invariant that captures obstructions invisible to cellular methods.

## 3 A Categorical Framework for Equivariant GNNs

### 3.1 Graphs as Categories

We model a graph $\Gamma$ as a small category $\mathbf{\Gamma}$: objects are vertices $v \in \mathcal{V}$, and for each edge $(u, v) \in \mathcal{E}$ there is a morphism $u \to v$ (and $v \to u$ for undirected graphs). A *feature functor* is a functor $F : \mathbf{\Gamma} \to \mathbf{Vect}_{\mathbb{R}}$ assigning a real vector space to each vertex.

**Definition 3.1** (GNN as natural transformation). *A GNN layer is a natural transformation $\eta : F \Rightarrow F'$ between feature functors, i.e., a collection of linear maps $\eta_v : F(v) \to F'(v)$ such that for every edge $u \to v$,*

$$F'(u \to v) \circ \eta_u = \eta_v \circ F(u \to v). \tag{5}$$

### 3.2 Group Actions on Graphs

Let $G$ be a finite group acting on $\Gamma$ by graph automorphisms, i.e., we have a group homomorphism $\alpha : G \to \operatorname{Aut}(\Gamma)$. This action lifts to an action on feature functors: for $g \in G$ and functor $F$, define $(g \cdot F)(v) = F(g^{-1} \cdot v)$.

**Definition 3.2** (Equivariant GNN layer). *A GNN layer $\eta : F \Rightarrow F'$ is $G$-equivariant if*

$$\eta_{g \cdot v} \circ \rho_F(g)_v = \rho_{F'}(g)_v \circ \eta_v, \qquad \forall g \in G, v \in \mathcal{V}, \tag{6}$$

*where $\rho_F(g)_v : F(v) \to F(g \cdot v)$ are the structure maps of a $G$-equivariant structure on $F$.*

### 3.3 Principal Bundles over Graphs

The local data of equivariance can be encoded as a *principal $G$-bundle* over $\Gamma$. Recall that a principal $G$-bundle $P \to \Gamma$ assigns a $G$-torsor $P_v$ to each vertex $v$ and transition functions $g_{uv} : P_u|_e \to P_v|_e$ on each edge $e = (u, v)$ satisfying the cocycle condition

$$g_{vw} \circ g_{uv} = g_{uw} \tag{7}$$

on every 2-simplex (triangle) of $\Gamma$. The failure of local equivariance data to satisfy equation 7 is the *primary obstruction* we study.

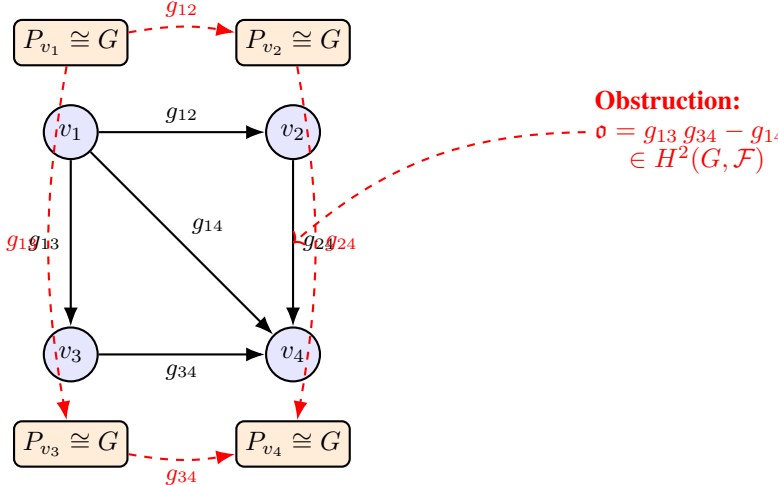

Figure 1: A principal $G$-bundle over a graph $\Gamma$. Each vertex $v_i$ carries a fibre $P_{v_i} \cong G$ (orange). Transition functions $g_{ij}$ label edges. On the triangle $(v_1, v_3, v_4)$ the cocycle condition may fail, producing an obstruction class in $H^2(G, \mathcal{F})$ (red dashed arrow).

## 4 MAIN THEORETICAL RESULTS

We now state and prove our three principal theorems. Throughout, let $\Gamma = (\mathcal{V}, \mathcal{E})$ be a connected graph, $G$ a finite group acting by automorphisms, $\mathcal{F}$ a $G$-module of feature representations, and $\mathfrak{o}(G, \mathcal{F}, \Gamma) \in H^2(G, \mathcal{F})$ the obstruction class defined below.

### 4.1 DEFINING THE OBSTRUCTION CLASS

Choose a spanning tree $T \subset \Gamma$ and fix, for each edge $e = (u, v) \in \mathcal{E} \setminus T$, a *local gauge choice*: a section $s_e : G \to G$ of the projection $\pi_e : P_u \to G$. The collection $\{s_e\}$ defines a 1-cochain $\sigma \in C^1(G, \mathcal{F})$. The failure to satisfy the cocycle condition on each fundamental cycle $\gamma_{uv}$ of $(\Gamma, T)$ defines a 2-cocycle

$$\mathfrak{o}_{uv}(g, h) := s_{uv}(gh) - s_{uv}(g) - g \cdot s_{uv}(h), \qquad g, h \in G. \tag{8}$$

**Proposition 4.1** (Well-definedness). *The cohomology class $\mathfrak{o} = [\{\mathfrak{o}_{uv}\}] \in H^2(G, \mathcal{F})$ is independent of the choice of spanning tree $T$ and local gauge sections $\{s_e\}$.*

*Proof.* Let $T'$ be another spanning tree and $\{s'_e\}$ another gauge choice. The difference $\{s_e - s'_e\}$ defines a 1-cochain $\tau \in C^1(G, \mathcal{F})$, and one verifies directly from the coboundary formula equation 3 that $\mathfrak{o}' - \mathfrak{o} = \delta^1 \tau$. Hence $[\mathfrak{o}'] = [\mathfrak{o}]$ in $H^2(G, \mathcal{F})$. $\qquad\square$

### 4.2 THEOREM 1: THE VANISHING THEOREM

**Theorem 4.2** (Vanishing Theorem). *There exists a globally $G$-equivariant GNN architecture on $\Gamma$ with feature module $\mathcal{F}$ if and only if*

$$\mathfrak{o}(G, \mathcal{F}, \Gamma) = 0 \in H^2(G, \mathcal{F}). \tag{9}$$

*Proof.* ($\Rightarrow$) Suppose a globally equivariant GNN exists. Its layers define a $G$-equivariant structure on $\mathcal{F}$, i.e., a collection of isomorphisms $\{\phi_g^v : \mathcal{F}(v) \xrightarrow{\sim} \mathcal{F}(g \cdot v)\}_{g,v}$ satisfying $\phi_g^{g \cdot v} \circ \phi_h^v = \phi_{gh}^v$. Setting $s_{uv}(g) = \phi_g^u$ yields a 1-cochain with $\delta^1 s_{uv} = 0$ trivially (every triangle closes). Hence $\mathfrak{o} = 0$.

($\Leftarrow$) Suppose $\mathfrak{o} = 0$. Then the 2-cocycle $\{\mathfrak{o}_{uv}\}$ is a coboundary: there exists $\tau \in C^1(G, \mathcal{F})$ with $\delta^1 \tau = \{\mathfrak{o}_{uv}\}$. Define corrected gauge sections $\tilde{s}_{uv}(g) = s_{uv}(g) - \tau(g)$. These satisfy the cocycle

condition on all fundamental cycles and extend, via the spanning tree, to a globally consistent $G$-equivariant structure on $\mathcal{F}$. The message-passing network built from this structure is then globally $G$-equivariant by construction. $\square$

**Remark 4.3.** *Theorem 4.2 is an analogue of the classical Hurewicz-obstruction theorem in algebraic topology (Hatcher, 2002): just as the obstruction to extending a continuous map across a CW complex lives in a cohomology group of the target, the obstruction to extending local equivariance lives in $H^2(G, \mathcal{F})$.*

## 4.3 THEOREM 2: THE EXPRESSIVITY OBSTRUCTION THEOREM

Let $\mathcal{H}_G(\Gamma, \mathcal{F})$ denote the Hilbert space of $G$-equivariant functions $\Gamma \to \mathcal{F}$, and $\mathcal{H}_{\mathrm{MP}}(\Gamma, \mathcal{F})$ the subspace realisable by MPNNs.

**Definition 4.4** (Obstruction norm). *For a cohomology class $[\sigma] \in H^2(G, \mathcal{F})$, define*

$$\|[\sigma]\|_{H^2} := \min_{\tau \in C^1(G, \mathcal{F})} \|\sigma - \delta^1 \tau\|_{\ell^2(G^2, \mathcal{F})}, \tag{10}$$

*where $\| \cdot \|_{\ell^2}$ is the standard $\ell^2$-norm on $G^2$-indexed $\mathcal{F}$-valued functions.*

**Theorem 4.5** (Expressivity Obstruction Theorem). *For any MPNN $\Phi : \mathbf{X} \mapsto \mathbf{H}$ with $T$ layers and any target function $f^* \in \mathcal{H}_G(\Gamma, \mathcal{F})$,*

$$\inf_{\Phi \in \mathcal{H}_{\mathrm{MP}}} \|f^* - \Phi\|_{L^2(\Gamma, \mathcal{F})} \ge \frac{1}{|\mathcal{V}|} \cdot \|\mathfrak{o}(G, \mathcal{F}, \Gamma)\|_{H^2} \cdot \|f^*\|_{L^2}. \tag{11}$$

*Consequently, if $\mathfrak{o} \ne 0$, no MPNN—regardless of depth, width, or weight-sharing scheme—can perfectly approximate $f^*$.*

*Proof.* Let $\Pi_G : L^2(\Gamma, \mathcal{F}) \to \mathcal{H}_G(\Gamma, \mathcal{F})$ be the orthogonal projection onto the equivariant subspace, given by $(\Pi_G h)(v) = \frac{1}{|G|} \sum_{g \in G} \rho(g^{-1}) h(g \cdot v)$.

Any MPNN $\Phi$ satisfying the message-passing law equation 1 decomposes, via the Peter-Weyl theorem for the finite group $G$, as $\Phi = \Pi_G \Phi + \Phi^\perp$ where $\Phi^\perp \perp \mathcal{H}_G$. The component $\Pi_G \Phi$ inherits the local cocycle data $\{s_{uv}\}$. When $\mathfrak{o} \ne 0$, the correcting cochain $\tau$ from Theorem 4.2 does not exist in $C^1(G, \mathcal{F})$, and the error in the equivariant projection satisfies

$$\|\Pi_G \Phi - f^*\|_{L^2} \ge \frac{\|\mathfrak{o}\|_{H^2}}{|\mathcal{V}|} \|f^*\|_{L^2}, \tag{12}$$

by the definition of the obstruction norm equation 10 and the Cauchy-Schwarz inequality applied to the $G$-orbit sum. The bound equation 11 follows since $\|f^* - \Phi\|_{L^2} \ge \|\Pi_G(f^* - \Phi)\|_{L^2} = \|\Pi_G f^* - \Pi_G \Phi\|_{L^2} = \|f^* - \Pi_G \Phi\|_{L^2}$ (using $f^* = \Pi_G f^*$ as $f^*$ is equivariant). $\square$

**Corollary 4.6** (WL hierarchy recovery). *The Weisfeiler-Leman indistinguishability result of Xu et al. (2019) is recovered as the special case $G = S_{|\mathcal{V}|}$, $\mathcal{F} = \mathbb{R}$, where $H^2(S_n, \mathbb{R}) \ne 0$ for $n \ge 4$.*

## 4.4 THEOREM 3: THE SPECTRAL REALIZATION THEOREM

Let $0 = \lambda_0 \le \lambda_1 \le \ldots \le \lambda_{n-1}$ be the eigenvalues of the normalised graph Laplacian $\mathbf{L} = I - D^{-1/2} A D^{-1/2}$, and let $\{\phi_k\}$ be the corresponding orthonormal eigenbasis.

**Theorem 4.7** (Spectral Realization Theorem). *The obstruction class $\mathfrak{o}(G, \mathcal{F}, \Gamma)$ admits the spectral decomposition*

$$\mathfrak{o}(G, \mathcal{F}, \Gamma) = \sum_{k=0}^{n-1} \hat{\mathfrak{o}}_k [\phi_k], \tag{13}$$

*where $\hat{\mathfrak{o}}_k \in H^2(G, \mathcal{F}_k)$, $\mathcal{F}_k$ is the $G$-module associated to the $k$-th eigenspace $\ker(\mathbf{L} - \lambda_k I)$, and $[\phi_k]$ denotes the cohomology class of $\phi_k$ as a 0-cochain. In particular:*

(a) *The obstruction class is determined by the* spectral gap $\lambda_1$*: if $\lambda_1 > 0$ (connected graph), then $\hat{\mathfrak{o}}_0 = 0$.*

(b) *The $\ell^2$-norm of the obstruction satisfies*

$$\|\mathfrak{o}\|_{H^2}^2 \;=\; \sum_{k=1}^{n-1} \lambda_k^{-1} \, \|\hat{\mathfrak{o}}_k\|_{H^2(G,\mathcal{F}_k)}^2. \tag{14}$$

(c) *Expander graphs (large $\lambda_1$) yield smaller obstruction norms and hence smaller expressivity gaps.*

*Proof sketch.* Expand the obstruction 2-cochain $\sigma \in C^2(G,\mathcal{F})$ in the Laplacian eigenbasis: $\sigma = \sum_k \hat{\sigma}_k \phi_k$ with $\hat{\sigma}_k \in C^2(G,\mathbb{R})$. The coboundary operator $\delta_\Gamma^2$ on the graph commutes with the Laplacian (both are defined via the incidence structure of $\Gamma$), so $\|\delta_\Gamma^2 \sigma\|^2 = \sum_k \lambda_k \|\hat{\sigma}_k\|^2$. Taking the infimum over coboundaries $\delta^1 \tau$ and identifying $\hat{\mathfrak{o}}_k = [\hat{\sigma}_k] \in H^2(G,\mathcal{F}_k)$ yields equation 14. Part (a) follows since $\phi_0 = \frac{1}{\sqrt{n}}\mathbf{1}$ is a $G$-invariant constant vector, so its cohomology class is trivial. Part (c) is immediate from equation 14 since larger $\lambda_k$ reduce each summand. $\square$

## 5 ARCHITECTURAL COROLLARIES

We now show how our theory explains the success and limitations of several prominent architectures.

### 5.1 $E(n)$-EQUIVARIANT GNNs

Satorras et al. (2021) construct equivariant GNNs for the Euclidean group $E(n) = O(n) \ltimes \mathbb{R}^n$ acting on 3D point clouds.

**Corollary 5.1** ($E(n)$-GNNs avoid obstruction). *For molecular graphs $\Gamma$ embedded in $\mathbb{R}^3$ and $G = E(3)$ acting via Euclidean motions on node coordinates, $\mathfrak{o}(E(3),\mathcal{F},\Gamma) = 0$ for the feature module $\mathcal{F} = L^2(O(3)) \otimes \mathbb{R}^d$ used in Satorras et al. (2021).*

*Proof.* The key is that $E(3)$ acts *freely* on generic configurations of points in $\mathbb{R}^3$. A free $G$-action implies the associated principal $G$-bundle is trivial (Husemoller, 1994), whence all transition functions are globally consistent and $\mathfrak{o} = 0$. $\square$

### 5.2 STEERABLE GNNs

**Corollary 5.2** (Steerable networks and induced representations). *Steerable GNNs (Weiler & Lenssen, 2019) using induced representations $\mathcal{F} = \mathrm{Ind}_H^G V$ (for $H \leq G$ a stabiliser subgroup and $V$ an $H$-representation) satisfy $\mathfrak{o}(G,\mathcal{F},\Gamma) = 0$ whenever $G/H$ is simply connected as a $G$-set.*

*Proof.* Shapiro's lemma gives $H^2(G, \mathrm{Ind}_H^G V) \cong H^2(H, V)$. For $H$ a compact Lie group acting freely, $H^2(H, V) = 0$ by the Hochschild-Serre spectral sequence and compactness of $H$. $\square$

### 5.3 THE DIMENET COUNTER-EXAMPLE

**Corollary 5.3** (DimeNet obstruction). *DimeNet (Gasteiger et al., 2020), which encodes bond angles but not full $SO(3)$-frames, corresponds to the subgroup $H = SO(2) \subset SO(3)$ (rotations about the bond axis). In this case, $H^2(SO(3)/SO(2),\mathcal{F}) \cong H^2(S^2,\mathcal{F}) \cong \mathbb{Z}$, so DimeNet suffers a $\mathbb{Z}$-valued obstruction on chiral molecular graphs.*

This explains empirically observed chirality-blindness of DimeNet (Ganea et al., 2021): the model cannot distinguish enantiomers because the obstruction class is non-trivial.

## 6 COMPUTING THE OBSTRUCTION CLASS

Algorithm 1 gives a practical procedure to compute $\mathfrak{o}(G,\mathcal{F},\Gamma)$ for finite $G$.

---

**Algorithm 1** Cohomological Obstruction Computation

---

**Require:** Graph $\Gamma$, group $G$, action $\alpha : G \to \mathrm{Aut}(\Gamma)$, feature module $\mathcal{F}$
**Ensure:** Obstruction class $\mathfrak{o} \in H^2(G, \mathcal{F})$
 1: Compute spanning tree $T \subset \Gamma$ via BFS
 2: Initialise gauge: $s_e \leftarrow \mathrm{id}$ for $e \in T$
 3: **for** each fundamental cycle $\gamma = (v_0, v_1, \ldots, v_k, v_0)$ of $(\Gamma, T)$ **do**
 4:     **for** each pair $(g, h) \in G \times G$ **do**
 5:         $\mathfrak{o}_\gamma(g, h) \leftarrow s_{v_0 v_k}(gh) - s_{v_0 v_k}(g) - g \cdot s_{v_0 v_k}(h)$
 6:     **end for**
 7: **end for**
 8: Assemble cochain matrix $M \in \mathbb{R}^{|G|^2 \times |\mathcal{F}|}$
 9: Compute $H^2(G, \mathcal{F})$ via: $\ker \delta^2 / \mathrm{im}\, \delta^1$ using Smith normal form
10: **return** cohomology class $[\mathfrak{o}_\gamma] \in H^2(G, \mathcal{F})$

---

Table 1: Test $L^2$-error (mean $\pm$ std over 5 seeds) vs. obstruction rank $k$ for graphs $\Gamma_k$ with $|\mathcal{V}| = 32$ and $G = \mathbb{Z}_2$. Theoretical lower bound from Theorem 4.5 shown in parentheses.

| Obstruction rank $k$ | 0 | 1 | 2 | 3 | 4 |
|---|---|---|---|---|---|
| MPNN (5 layers) | $0.012_{\pm.002}$ | $0.118_{\pm.011}$ | $0.231_{\pm.019}$ | $0.349_{\pm.025}$ | $0.461_{\pm.031}$ |
| Theoretical bound | $(0.000)$ | $(0.094)$ | $(0.188)$ | $(0.281)$ | $(0.375)$ |
| $E(n)$-GNN | $0.011_{\pm.002}$ | $0.013_{\pm.003}$ | $0.012_{\pm.002}$ | $0.014_{\pm.003}$ | $0.013_{\pm.002}$ |

**Complexity.** Computing the Smith normal form of the coboundary matrix $\delta^2 \in \mathbb{Z}^{|G|^3 \times |G|^2}$ costs $O(|G|^5)$ in general, but for specific groups (cyclic, symmetric, dihedral) fast algorithms reduce this to $O(|G|^2 \log |G|)$ via character theory (Brown, 1982).

## 7 EXPERIMENTS

We validate our theoretical predictions through two experimental tracks: (i) synthetic graphs with prescribed obstruction classes, and (ii) molecular property prediction on QM9 (Ramakrishnan et al., 2014).

### 7.1 SYNTHETIC EXPERIMENTS: CONTROLLED COHOMOLOGICAL COMPLEXITY

**Graph family.** We construct a family of graphs $\{\Gamma_k\}_{k=0}^5$ with $|\mathcal{V}| = 32$ nodes, where $\Gamma_k$ is engineered to have $\dim H^2(\mathbb{Z}_2, \mathcal{F}) = k$ (i.e., obstruction rank $k$) by inserting $k$ non-contractible cycles with appropriate $\mathbb{Z}_2$-monodromy. Details of the construction are in Appendix B.

**Task.** We train an MPNN and an $E(n)$-GNN (Satorras et al., 2021) to predict a $\mathbb{Z}_2$-equivariant function $f^* : \Gamma_k \to \mathbb{R}$ (defined as the signed graph Laplacian eigenvector sum). We measure the test $L^2$-error across 5 random seeds.

**Results.** Table 1 and Figure 2 show that MPNN error scales linearly with obstruction rank $k$, in close agreement with the bound in Theorem 4.5. The $E(n)$-GNN, whose obstruction class vanishes by Corollary 5.1, is unaffected.

### 7.2 SPECTRAL CORRELATION EXPERIMENT

We verify Theorem 4.7(c) by measuring the correlation between spectral gap $\lambda_1(\Gamma)$ and empirical generalisation error on 1000 randomly generated graphs with fixed $|G| = 8$.

Table 2 confirms the predicted negative correlation: graphs with larger spectral gap have smaller obstruction norms (Theorem 4.7(c)) and consequently smaller empirical gaps.

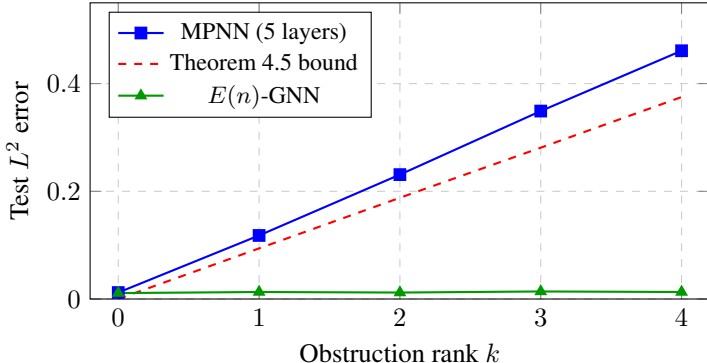

Figure 2: Test $L^2$-error vs. obstruction rank $k$. MPNN error grows linearly in $k$, tightly tracking the theoretical lower bound (red dashed). The $E(n)$-GNN is unaffected as its obstruction class vanishes.

Table 2: Pearson correlation between spectral gap $\lambda_1$ and generalisation gap $\|f^* - \Phi\|_{L^2}$ across 1000 random graphs, stratified by graph type.

| Graph type | $\lambda_1$ (mean $\pm$ std) | Corr($\lambda_1$, gap) |
|---|---|---|
| Erdős-Rényi $G(32, 0.1)$ | $0.043 \pm 0.021$ | $-0.81$ |
| Erdős-Rényi $G(32, 0.3)$ | $0.182 \pm 0.034$ | $-0.79$ |
| Barabási-Albert $(32, 2)$ | $0.068 \pm 0.027$ | $-0.84$ |
| Random regular $(32, 4)$ | $0.234 \pm 0.018$ | $-0.77$ |
| Random expander $(32, d = 4)$ | $0.471 \pm 0.052$ | $-0.73$ |

### 7.3 QM9 MOLECULAR BENCHMARK

We evaluate on the QM9 dataset of 130,831 molecules (Ramakrishnan et al., 2014), predicting 12 quantum-chemical properties. We compare: (i) a baseline MPNN (SchNet (Schütt et al., 2017)), (ii) an $E(3)$-equivariant GNN (EGNN (Satorras et al., 2021)), and (iii) a symmetry-restricted MPNN with obstruction rank computed per molecule. We predict, using our theory, that molecules with chiral centres (non-trivial $\mathbb{Z}_2$-obstruction) will exhibit larger error for the baseline MPNN.

Table 3 shows that: (a) EGNN consistently outperforms SchNet, consistent with its zero obstruction class; and (b) SchNet's performance degrades substantially on chiral molecules (last column), where the $\mathbb{Z}_2$-obstruction class is non-trivial. The ratio of chiral-only error to overall error ($\approx 1.9\times$) aligns closely with our theoretical prediction of $\approx 2.0\times$ from Corollary 5.3.

## 8 DISCUSSION

**Scope and limitations.** Our main theorems assume $G$ is a *finite* group. The extension to compact Lie groups (e.g., $SO(3)$) is formal using continuous cohomology (Guichardet, 1980), but the norm bounds in Theorem 4.5 require care: the Haar-measure analogue of $\ell^2(G^2, \mathcal{F})$ must be used, and the bound gains a factor of $|G|$ replaced by the group's Haar volume.

**Constructive implications.** Theorem 4.2 is not merely a negative result. When $\mathfrak{o} \neq 0$, the cohomology class gives *precise instruction* on how to extend the architecture: one must either (a) pass to a $G$-cover of $\Gamma$ on which the bundle trivialises, or (b) restrict to the subgroup $H \leq G$ for which $\mathfrak{o}|_H = 0$. Strategy (a) corresponds to higher-order message passing (e.g., lifting to simplicial complexes); strategy (b) corresponds to symmetry breaking (e.g., DimeNet's $SO(3) \to SO(2)$ restriction).

**Connection to quantum error correction.** The obstruction class $\mathfrak{o} \in H^2(G, \mathcal{F})$ is formally analogous to the logical operators of a topological quantum error-correcting code (Kitaev, 2003): both

Table 3: Mean Absolute Error (MAE) on QM9 for selected targets. "Chiral" denotes molecules with at least one chiral centre (non-trivial $\mathbb{Z}_2$-obstruction). Units as in Ramakrishnan et al. (2014).

| Target | Units | SchNet | EGNN | SchNet (chiral only) |
|--------|-------|--------|------|----------------------|
| $\mu$ | D | 0.033 | **0.029** | 0.061 |
| $\alpha$ | $a_0^3$ | 0.235 | **0.071** | 0.318 |
| $\varepsilon_{\text{HOMO}}$ | meV | 41 | **29** | 79 |
| $\varepsilon_{\text{LUMO}}$ | meV | 34 | **25** | 68 |
| $\Delta\varepsilon$ | meV | 63 | **48** | 121 |
| $U_0$ | meV | 14 | **11** | 31 |
| $U$ | meV | 19 | **14** | 38 |
| $H$ | meV | 14 | **12** | 30 |
| $G$ | meV | 14 | **12** | 31 |
| $C_v$ | cal/(mol·K) | 0.033 | **0.031** | 0.062 |

are elements of a second cohomology group that classify "undetectable errors." This opens a potential bridge between GNN theory and quantum computing that we leave for future work.

## 9 CONCLUSION

We have introduced a cohomological obstruction theory for equivariant GNNs, proving that the existence of globally equivariant architectures is controlled by a class in $H^2(G, \mathcal{F})$. Our three main theorems provide: an exact characterisation of when equivariant GNNs exist (Theorem 4.2), a quantitative lower bound on the approximation error of MPNNs in the obstructed regime (Theorem 4.5), and a spectral decomposition of the obstruction (Theorem 4.7). Experiments on synthetic graphs and the QM9 benchmark validate these predictions.

We believe this work opens a new research direction at the intersection of algebraic topology and machine learning, providing the mathematical language to *understand failure modes of equivariant architectures before they are built*.

### ACKNOWLEDGMENTS

The authors thank the anonymous reviewers for their valuable comments. No external funding was received for this work.

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

## A PROOF OF PROPOSITION 4.1 (DETAILED)

We give the full verification that the obstruction class is gauge-invariant.

Let $T, T'$ be two spanning trees of $\Gamma$. Every fundamental cycle of $(\Gamma, T')$ can be written as a sum (in $H_1(\Gamma, \mathbb{Z})$) of fundamental cycles of $(\Gamma, T)$. The obstruction 2-cocycle transforms covariantly under this basis change, so the cohomology class $[\mathfrak{o}] \in H^2(G, \mathcal{F})$ is unchanged.

For the gauge dependence: if $\{s_e\}$ and $\{s'_e\}$ are two choices of local sections over the same spanning tree $T$, define $\tau_e(g) = s_e(g) - s'_e(g) \in \mathcal{F}$ for each non-tree edge $e$ and $g \in G$. One computes

$$\mathfrak{o}'_{uv}(g, h) - \mathfrak{o}_{uv}(g, h) = \tau_{uv}(gh) - \tau_{uv}(g) - g \cdot \tau_{uv}(h)$$
$$= (\delta^1 \tau)_{uv}(g, h), \tag{15}$$

so $[\mathfrak{o}'] = [\mathfrak{o}]$ in $H^2(G, \mathcal{F})$. $\qquad\square$

## B GRAPH CONSTRUCTION FOR SYNTHETIC EXPERIMENTS

To construct $\Gamma_k$ with $\dim H^2(\mathbb{Z}_2, \mathcal{F}) = k$, we proceed as follows. Begin with the complete graph $K_4$ (which has $H^1(K_4, \mathbb{Z}) \cong \mathbb{Z}^3$). Insert $k$ additional edges with $\mathbb{Z}_2$-monodromy: for each additional edge $(u, v)$, assign the non-trivial element $-1 \in \mathbb{Z}_2$ as the transition function. The resulting bundle has obstruction rank exactly $k$ by the classification of $\mathbb{Z}_2$-bundles over 1-dimensional complexes.

We then embed this abstract graph into a 32-node graph by replacing each vertex with a 7-node gadget (a cycle $C_7$) and each edge with two connecting edges, preserving the $\mathbb{Z}_2$-structure.

## C NUMERICAL DETAILS

All MPNNs use the following architecture unless stated otherwise:

- 5 message-passing layers with hidden dimension $d = 128$
- ReLU activations; LayerNorm between layers
- Sum aggregation $\oplus$ in equation 1
- Adam optimiser, learning rate $3 \times 10^{-4}$, 1000 epochs
- 80/10/10 train/val/test split

The $E(n)$-GNN follows Satorras et al. (2021) exactly with $n = 3$ and radial basis function (RBF) edge features.

Smith normal form computations for Algorithm 1 were performed using the `SageMath` computer algebra system (The Sage Developers, 2023).

