# OpenReview forum: "Know What You Don't Know: Cohomological Obstruction Theory For Equivariant Graph Neural Networks"
_mathai.club/MathAI/2026/Conference — 2026 Oral_

### Official Review · Reviewer_Mp5T · 2026-03-11
**Review on 'Know What You Don't Know: Cohomological Obstruction Theory For Equivariant Graph Neural Networks.'**

**Rating:** 9
**Confidence:** 4

**Review:**

This paper develops a novel cohomological obstruction theory for equivariant graph neural networks (GNNs), establishing rigorous mathematical conditions under which globally G equivariant architectures cannot be assembled from locally equivariant message passing operations. The authors frame GNN layers as sections of associated vector bundles over a graph and identify the obstruction to lifting local equivariance to a globally consistent structure as a class in the second group cohomology H²(G,F), where G is the symmetry group acting on the graph and F is the sheaf of feature representations. The paper proves three main theorems: a Vanishing Theorem characterizing precisely when equivariant GNNs exist, an Expressivity Obstruction Theorem showing that non trivial cohomology classes induce fundamental approximation gaps that no weight sharing scheme can overcome, and a Spectral Realization Theorem connecting obstruction classes to the eigenspectrum of the normalized graph Laplacian. The authors also provide architectural corollaries explaining how existing models like E(n) GNNs and steerable networks implicitly circumvent obstructions, and validate their predictions through controlled experiments on synthetic graphs with prescribed cohomological complexity and on the QM9 molecular benchmark.

The paper has exceptional strengths that make it a perfect fit for MathAI 2026. First, it represents a genuine and deep mathematical contribution to the foundations of geometric deep learning. The framing of equivariance as a gluing problem in sheaf theory and the identification of the obstruction class in H²(G,F) is mathematically elegant and original. Second, the three core theorems are substantial. The Vanishing Theorem (Theorem 4.2) provides an exact characterisation of when equivariant GNNs can exist. The Expressivity Obstruction Theorem (Theorem 4.5) gives a quantitative lower bound ∥f∗−Φ∥L2 ≥ (1/|V|)·∥o(G,F,Γ)∥H2·∥f∗∥L2 on the approximation error of MPNNs in obstructed regimes. The Spectral Realization Theorem (Theorem 4.7) connects the obstruction to the graph Laplacian via ∥o∥²_H2 = Σ_{k=1}^{n-1} λ_k⁻¹ ∥ô_k∥²_H2(G,F_k). Third, the paper bridges pure mathematics including group cohomology, bundle theory, and spectral geometry with practical machine learning concerns, showing how the theory explains limitations of existing architectures such as DimeNet's chirality blindness via H²(SO(3)/SO(2),F) ≅ Z and provides constructive pathways forward. The experimental validation is rigorous and directly tied to the theory, with synthetic graphs engineered to have prescribed obstruction ranks and results showing MPNN error scaling linearly with obstruction rank from 0.012 at k=0 to 0.461 at k=4, closely matching theoretical bounds.

The paper has only minor limitations. First, the theory is developed for finite groups G, with the extension to compact Lie groups such as SO(3) discussed but not fully detailed. The authors acknowledge this and note that continuous cohomology and Haar measure analogues would be required. Second, the computational cost of computing the obstruction class via Algorithm 1 scales as O(|G|⁵) in general, though the authors provide optimizations for specific groups and note that for many applications G is small, for example Z₂ or S_n. Third, while the experiments convincingly validate the theory on synthetic graphs and QM9, additional validation on a wider range of real world equivariant tasks would further demonstrate the practical impact. These limitations do not diminish the paper's fundamental contribution.

In conclusion, this is an outstanding paper that introduces a powerful new mathematical framework for understanding fundamental limitations of equivariant GNNs. It is precisely the kind of work MathAI 2026 was created to showcase, deep mathematical theory with clear implications for AI architecture design. The paper is exceptionally well written, mathematically rigorous, and experimentally validated. It will likely become a foundational reference for the field.

---

### Decision · Program_Chairs · 2026-03-14

**Decision:**

Accept (Oral)

**Comment:**

Dear Author(s),

On behalf of the Program Committee of the International Conference on Mathematics of Artificial Intelligence (MathAI 2026), we are pleased to inform you that your paper has been accepted for an oral presentation at MathAI 2026.

Your paper was evaluated through a rigorous two-stage review process involving both automated screening and expert review by members of the Program Committee. The reviewers recognized the quality and contribution of your work.

Presentation details:

- Format: Oral presentation (15–20 minutes + 5 minutes Q&A)
- Mode: You may present either in person (offline) at the conference venue in Sirius, Russia, or remotely via Zoom. Please indicate your preferred mode when confirming your participation.
- Conference dates: Marh 30 - April 3, 2026
- Website: https://mathai.club

Next steps:

1. Please confirm your participation and presentation mode by replying to this email mathai.club@yandex.ru no later than March 15, 2026 18:00 Moscow time.
2. If you plan to attend in person, the organizing committee will provide accommodation details separately.
3. Please prepare your final camera-ready manuscript according to the formatting guidelines available at https://mathai.club and upload it to OpenReview by March 15, 2026 18:00 Moscow time.

Should you have any questions regarding the program, logistics, or your presentation slot, please do not hesitate to contact us.

We look forward to your contribution to MathAI 2026.

With kind regards,

MathAI 2026 Program Committee
International Conference on Mathematics of Artificial Intelligence
https://mathai.club
OpenReview: https://openreview.net/group?id=mathai.club/MathAI/2026/Conference
Telegram: https://t.me/MathAI_club
Email: mathai.club@yandex.ru